# Accelerating Training of Deep Spiking Neural Networks with Parameter Initialization

## Abstract

Although spiking neural networks (SNNs) show strong advantages in terms of information encoding, power consumption, and computational capability, the underdevelopment of supervised learning algorithms is still a hindrance for training SNNs. We consider weight initialization to be a pivotal issue for efficient SNN training. It greatly influences gradient generation when using the method of backpropagation through time in the initial training stage. Focusing on the properties of spiking neurons, we first derive an asymptotic formula for their response curve, approximating the actual neuron response distribution. Then, we propose an initialization method based on the slant asymptote to overcome gradient vanishing. Finally, experiments with different coding schemes in classification tasks on the MNIST and CIFAR10 dataset show that our method can effectively improve the training speed and the model accuracy compared with traditional deep learning initialization methods and existing SNN initialization methods. Further validation on different neuron types and training hyperparameters shows comparably good versatility and superiority over other methods. Some suggestions are given for SNN training based on the analyses.

## 1 Introduction

To date, deep learning has contributed to outstanding performance on various tasks (Hinton et al., 2012; Hosu & Rebedea, 2016; He et al., 2016; LeCun et al., 2015). Most state-of-the-art models are based on analog neurons. Each analog value conveys activation; thus, the inference processes of analog neural networks (ANNs) consist of massive matrix manipulations. To overcome this, spiking neural networks (SNNs) are increasing attracting from researchers (Maass, 1997; Maass & Markram, 2004; Yang et al., 2019). The spiking neurons in an SNN emit sparse 0,1 spikes when activated. Compared with existing ANNs, SNNs are considered to encode spatiotemporal patterns and have better power efficiency. Hence, implementations of SNNs are highly appealing in power-limited scenarios. In addition, SNNs have exhibited computational capabilities as powerful as those of conventional artificial neural networks (Lee et al., 2016).

Despite SNNs' powerful capabilities, these networks lack efficient supervised learning algorithms. The discrete spikes prevent the direct use of backpropagation (BP) training. Recently proposed supervised algorithms have mainly focused on two methodologies: ANN-SNN conversion and BP through time (BPTT) with surrogate functions. Compared to ANN-SNN conversion (Diehl et al., 2015; Sengupta et al., 2019; Deng & Gu, 2021; Ho & Chang, 2020; Ding et al., 2021), BPTT is more friendly to the event-based datasets (Lee et al., 2016; Zheng et al., 2021; Neftci et al., 2019). However, Wu et al. (2018) has reported that BPTT is only feasible to shallow architectures. Therefore, BPTT still cannot be regarded as a mature SNN training algorithm.

We ascribe the immaturity of BPTT training to the influence of the gradients. One common reason is gradient vanishing and explosion. This is natural since SNNs can be regarded as a type of recurrent neural networks (RNNs) (He et al., 2020; Fang et al., 2020). In the wisdom of RNN methodology, researchers have managed to provide a simple solution by rethinking weight initialization (Le et al., 2015). The success of RNN training inspires consideration of whether there may also be a simple solution for SNNs. Another reason is early gradient generation, which has rarely been mentioned in the previous literature. Most surrogate functions produce gradients when the membrane potential is in the neighborhood of the threshold. Some produce gradients only when spikes are fired (Lee

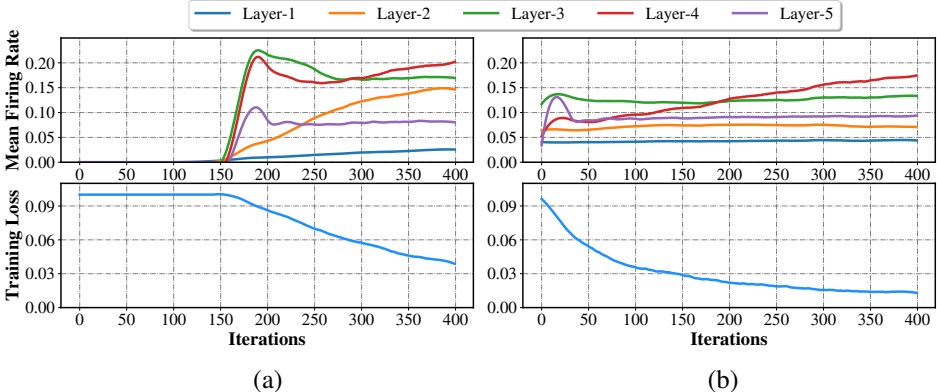

Figure 1: Loss curves and mean firing rate curves of a 5-layer SNN, initialized using the methods proposed by Glorot & Bengio (2010) (a) and us (b). Figure (a) shows an example of inadequate initialization. All losses start to decrease as the firing rates increase.

et al., 2020a). Thus, the flow of the gradients strongly depends on the number of firing neurons and will influence the optimization of the network. A small experiment can help illustrate how early gradient generation affects training. As shown in Fig. 1, with an inadequate weight initialization method, the neurons receive inputs that are too small and cannot fire initially, leading to stall of the training process because gradients fail to be produced. Until some neurons fire, the loss does not begin to decrease, and the neurons must suffer a latter uncoordinated training schedule. Therefore, in general, good weight initialization is also important for efficient SNN training.

At present, initialization for SNN training is mainly achieved by following and adapting methods originally developed for ANNs. However, unlike the rectified linear unit (ReLU) activation function conventionally used in ANNs, spiking neurons respond very differently, and recent improvements in models have rarely focused on the unique properties of spiking neurons. In this paper, we investigate the theoretical response of spiking neurons to develop a better SNN initialization strategy. The main contributions of this paper are summarized as follows:

- We use iterative systems to model first-order integrate-and-fire neurons and theoretically investigate the response curve. A slant asymptote is derived to approximate the actual neuron response.
- We examine the difficulty of training deep SNNs and develop an initialization method based on the slant asymptote. The proposed initialization method relies on adequate scaled weights and nonzero biases to enable early gradient training.
- Experiments with rate coding and event coding in classification tasks show that our method can effectively improve the model training speed and accuracy. In addition, empirical analyses suggest that our asymptote initialization method offers versatility and superiority over a vast range of training hyperparameters, such as different neuron types, surrogate functions, and optimizers.

## 2  THEORETICAL ANALYSIS OF SNNS

### 2.1  SPIKING NEURONS AND SURROGATE FUNCTIONS

With the development of neuroscience, many neuron models have been proposed (Hodgkin & Huxley, 1952; Gerstner et al., 2014), commonly in the form of differential systems. However, in the field of computing, analyses are often based on first-order differential models because of computational efficiency. Typically, leaky and nonleaky integrate-and-fire (LIF and IF) models can be formulated in terms of the dynamics of the membrane potential $v$ as shown in Eq. 1(a) and (b).

$$
\begin{aligned}
\tau \frac{d\boldsymbol{v}(t)}{dt} &= -(\boldsymbol{v}(t) - v_{reset}) + R\boldsymbol{i}(t) \quad (a)\ \tau \text{ is time constant.} \\
\frac{d\boldsymbol{i}(t)}{dt} &= \frac{1}{C}\boldsymbol{i}(t) \quad\quad\quad\quad\quad\quad (b)
\end{aligned}
\tag{1}
$$

Table 1: Parameter setting examples for first-order linear integrate-and-fire models expressed in the form given in Eq. 2.

| Model | $k$ | $\lambda$ |
|---|---|---|
| LIF model in Eq. 1(a) (Abbott, 1999) | $1 - 1/\tau$ | $R/\tau$ |
| IF model in Eq. 1(b) (Abbott, 1999) | $1$ | $1/C$ |
| Diehl et al. (2015); Sengupta et al. (2019) | $1$ | $1$ |
| Wu et al. (2018); Zheng et al. (2021) | $\tau_{decay}$ | $1$ |
| Panda et al. (2020) | $\alpha$ | $1$ |

By utilizing the Euler method, we can convert these dynamic systems into an iterative expression that computers prefer (Eq. 2). Note that $\boldsymbol{v}_{reset}$ is usually 0 in the literature; hence, it is neglected in latter analyses. When an element of $\boldsymbol{v}^t$ exceeds the firing threshold $\theta$, the corresponding neuron will emit a spike of 1, and membrane potential will then be reset to 0. Without loss of generality, we use the following equations to describe the iterative systems for first-order integrate-and-fire neurons.

$$
\begin{aligned}
\boldsymbol{u}^t &= k\boldsymbol{v}^t + \lambda\boldsymbol{i}^t \\
\boldsymbol{o}^t &= \boldsymbol{u}^t > \theta \\
\boldsymbol{v}^{t+1} &= \boldsymbol{u}^t(1 - \boldsymbol{o}^t)
\end{aligned}
\tag{2}
$$

In Eq. 2, $k$ and $\lambda$ are hyperparameters that control the neuron type and firing pattern ($k \in (0, 1]$). When $k$ is set to 0, the iteration of the membrane potential over time is canceled, and the network is not a standard SNN. Many neuron models used in the existing literature on SNN training can be regarded as the model given in Eq. 2 with different settings of the two parameters. Table 1 illustrates some of these settings.

In Eq. 2, $\boldsymbol{o}^t = \boldsymbol{u}^t > \theta$ is the non-differentiable part. It can also be rewritten as $\boldsymbol{o}^t = H(\boldsymbol{u}^t - \theta)$ where $H(\cdot)$ is the Heaviside step function. $H(x) = 0$ when $x \le 0$, and $H(x) = 1$ when $x > 0$. The heaviside function has a gradient of 0 almost everywhere. Surrogate function methods first emerged in network quantization to cope with non-differentiable problems (Mairal, 2013; Hubara et al., 2016). Accordingly, four surrogate functions (sigmoid, tanh, hardtanh, arctan) are presented in this paper, parameterized by the parameter $a$, as listed in Table A3. These functions are the shifted and scaled version of the original functions, as shown in Fig. A7 in the appendix. The parameter $a$ can be adjusted to rescale the output gradient. The gradient functions are typically symmetric w.r.t. an input of zero. An exception has been proposed by Lee et al. (2020a), such that only the production of spikes can trigger nonzero gradients.

## 2.2 RESPONSE OF SPIKING NEURONS AND ASYMPTOTES

It is not easy to establish the relationship between the current input and the output based on an iterative equation. Considering that the difference between the iterative equation and the differential equation is small when $stepsize = 1$, and that the differential equation can be integrated to obtain the response curve, Theorem 1 gives the response of a first-order integrate-and-fire neurons.

**Theorem 1.** *For a first-order integrate-and-fire neuron as expressed in Eq. 2 ($u^t = kv^t + \lambda i^t, k \in (0, 1]$), the model takes a constant current $i^t = i$ as input and produces spikes as output. Let the discrete output firing rate w.r.t. $i$ be expressed as $f = \frac{T}{N}$, where $T$ is the total inference time and $N$ is the spike count. Then, the input-output response curve ($f$-$i$ curve) is estimated as follows:*

$$
\begin{aligned}
&\text{If } k = 1: \\
&\quad f = \begin{cases} 0, & i \le 0, \\ \frac{\lambda}{\theta}i, & 0 < i < \frac{\theta}{\lambda}, \\ 1, & i \ge \frac{\theta}{\lambda}. \end{cases} \\
&\text{If } k \in (0, 1): \\
&\quad f = \begin{cases} 0, & i \le 0, \\ (k-1)/\ln(1 + \frac{(k-1)\theta}{\lambda i}), & 0 < i < \frac{\theta}{\lambda} \cdot \frac{k-1}{e^{k-1}-1}, \\ 1, & i \ge \frac{\theta}{\lambda} \cdot \frac{k-1}{e^{k-1}-1}. \end{cases}
\end{aligned}
\tag{3}
$$

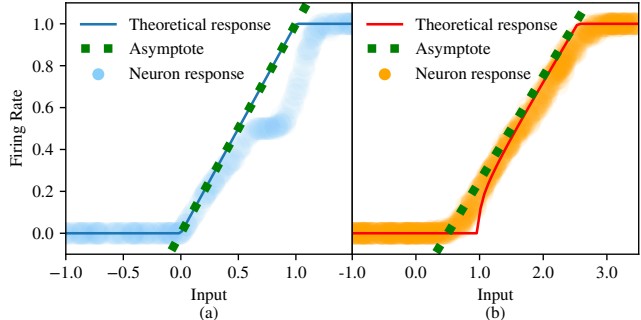

Figure 2: Theoretical response curves after integration (solid lines), slant asymptotes (dashed lines), and actual neuron responses (circular dots) for the first-order linear integrate-and-fire model with (a) $k = 1$ and (b) $k \in (0, 1)$.

The solid lines in Fig. 2 represent the response curves when $k = 1$ and $k \in (0, 1)$ in Theorem 1. Fig. 2 also presents scatter plots of the actual response of a spiking neuron given the corresponding parameters (marked with circular dots). The actual neuron responses match the theoretical $f - i$ curve for most inputs. Consider the region in which $f \in (0, 1)$. When $k \in (0, 1)$, it can be intuitively found that the $f$-$i$ curve, which has a non-differentiable point, seems to exhibit a slant asymptote. Theorem 2 manifests the existence of this slant asymptote and provides a uniform expression for it. In Fig. 2, this asymptote approximately crosses the actual neuron response distribution. In this case, the slant asymptote of $f$-$i$ can better describe the relationship between the expected input current and the output firing rate.

**Theorem 2.** *For a first-order integrate-and-fire neuron as expressed in Eq. 2 ($u^t = kv^t + \lambda i^t, k \in (0, 1]$), the slant asymptote of the $f$-$i$ curve satisfying $f \in (0, 1)$ exists and can be uniformly expressed as follows:*

$$f_a = \frac{\lambda}{\theta} i + \frac{k - 1}{2}. \tag{4}$$

## 3  WEIGHT INITIALIZATION FOR SNN TRAINING

The gradient of the weights is related to the spike output. To better illustrate this, given an $L$-layered SNN, we denote the input current, membrane potential, intermediate variable, and spike output of the $l^{th}$ layer by $\boldsymbol{i}^{l,t}, \boldsymbol{v}^{l,t}, \boldsymbol{u}^{l,t}$, and $\boldsymbol{o}^{l,t}$, respectively. The variables for the neurons in one layer satisfy Eq. 2. $\boldsymbol{i}^{l,t}$ is the weighted input consisting of the previous layer's spike output ($\boldsymbol{i}^{l,t} = \boldsymbol{W}^l \boldsymbol{o}^{l-1,t} + \boldsymbol{b}^l$), and the first layer takes the encoded spike $\boldsymbol{o}^{0,t}$ as input. When we backpropagate the loss $\varepsilon$ through time $T$, the gradients w.r.t. the weights of the $l^{th}$ layer can be expressed as shown in Eq. 5. An inappropriate $\boldsymbol{o}^{l-1,t}$, such as an output of zero, will lead to a poor gradient.

$$\frac{\partial \varepsilon}{\partial \boldsymbol{W}^l} = \sum_{t=1}^{T} \frac{\partial \varepsilon}{\partial \boldsymbol{u}^{l,t}} \frac{\partial \boldsymbol{u}^{l,t}}{\partial \boldsymbol{i}^{l,t}} \frac{\partial \boldsymbol{i}^{l,t}}{\partial \boldsymbol{W}^l} = \lambda \sum_{t=1}^{T} \frac{\partial \varepsilon}{\partial \boldsymbol{u}^{l,t}} (\boldsymbol{o}^{l-1,t})^{\mathrm{T}} \tag{5}$$

Therefore, the main concern is how to produce spikes to trigger the learning process. Thankfully, the clipped version of the slant asymptote can replace the analysis of complex iterative functions, allowing us to focus on the scale of the parameters:

$$\hat{\boldsymbol{i}}^l = \boldsymbol{W}^l \hat{\boldsymbol{o}}^{l-1} + \boldsymbol{b}^l$$
$$\hat{\boldsymbol{o}}^l = \mathrm{clip}(\frac{\lambda}{\theta} \hat{\boldsymbol{i}}^l + \frac{k-1}{2}, 0, 1) \tag{6}$$

where $\hat{\boldsymbol{i}}^i$ and $\hat{\boldsymbol{o}}^i$ are the expected current and firing rate vector, respectively, of the $i^{th}$ layer with $n_i$ neurons and $\boldsymbol{W}^i$ and $\boldsymbol{b}^i$ are the corresponding parameters. Note that $\boldsymbol{W}^i$ is an $n_i \times n_{i-1}$ matrix. $\lambda$, $k$ and $\theta$ represent the neuron parameters.

First, all weights in the SNN are initialized to be independent and identically distributed (i.i.d.). Let us assume that the input current variances are the same. Then, we have:

$$\mathrm{Var}(\hat{\boldsymbol{i}}^l) = n_{l-1} \mathrm{Var}(\boldsymbol{W}^l \hat{\boldsymbol{o}}^{l-1}) \tag{7}$$

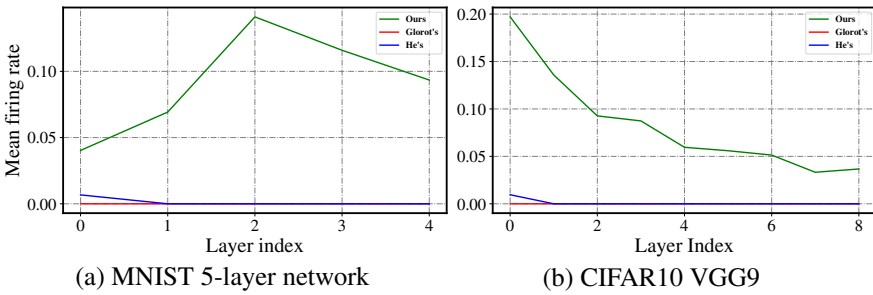

Figure 3: The illustration of the impact of initialization methods to the overall spike activity before training. The networks are randomly initialized using different initialization methods.

where $\hat{i}^l$, $\mathbf{W}^l$ and $\hat{o}^{l-1}$ denote random variables of elements in $\hat{i}^l$, $\mathbf{W}^l$ and $\hat{o}^{l-1}$, respectively. Considering $\text{Var}(\mathbf{W}^l \hat{o}^{l-1})$, $\text{Var}(\mathbf{W}^l \hat{o}^{l-1}) = \text{Var}(\mathbf{W}^l)\mathbb{E}[(\hat{o}^{l-1})^2] + \text{Var}(\hat{o}^{l-1})\mathbb{E}[\mathbf{W}^l]^2$, let $\mathbf{W}_l$ be of zero mean; then:

$$\text{Var}(\hat{i}^l) = n_{l-1}\text{Var}(\mathbf{W}^l)\mathbb{E}[(\hat{o}^{l-1})^2]. \tag{8}$$

In the function $\hat{o} = \text{clip}(\frac{\lambda}{\theta}\hat{i} + \frac{k-1}{2}, 0, 1)$, $i = \frac{\theta(1-k)}{2\lambda}$ is the transition point between the nonspiking region and the linear response region. When $b^l$ is initialized as $\frac{\theta(1-k)}{2\lambda}$ ($l = 1, 2, \cdots, L$), a zero-centered $\mathbf{W}^l \times \hat{o}^{l-1}$ produces a symmetric distribution around zero. With $b^l$ initialized as $\frac{\theta(1-k)}{2\lambda}$,

$$\mathbb{E}[(\hat{o}^{l-1})^2] = \frac{1}{2}\text{Var}(\frac{\lambda}{\theta}\hat{i}^{l-1}) = \frac{1}{2}\frac{\lambda^2}{\theta^2}\text{Var}(\hat{i}^{l-1}). \tag{9}$$

By plugging $\mathbb{E}[(\hat{o}^{l-1})^2]$ into Eq. 8, we can obtain:

$$\text{Var}(\hat{i}^l) = \frac{1}{2}\frac{\lambda^2}{\theta^2}n_{l-1}\text{Var}(\mathbf{W}^l)\text{Var}(\hat{i}^{l-1}) \tag{10}$$

To produce spikes is to maintain $\mathbb{E}[(\hat{o}^l)^2]$, which is related to $\text{Var}(\hat{i}^l)$ according to Eq. 9. Therefore, we determine the distribution of $\mathbf{W}^l$ as follows (for $l = 2, 3, \cdots, L$):

$$\frac{\text{Var}(\hat{i}^l)}{\text{Var}(\hat{i}^{l-1})} = \frac{1}{2}\frac{\lambda^2}{\theta^2}n_{l-1}\text{Var}(\mathbf{W}^l) = 1 \tag{11}$$

Accordingly, $\mathbf{W}^l$ is initialized from a Gaussian distribution $\mathcal{N}(0, \frac{2\theta^2}{\lambda^2 n_{l-1}})$, where $n_{l-1}$ is the number of input neurons in layer $l$. As done in other initializers, we apply the initialization to the first layer for simplicity.

In previous work on ANNs, there is no need for the neurons to accumulate input. Thus, the biases are often set to zero in an ANN. For SNNs, because the response asymptote indicates a nonzero non-differentiable point, it is natural for the biases in an SNN to be initialized to that non-differentiable value. We validate the impact to the mean firing rate of our initialization method in Fig. 3. Our method can effectively trigger spikes when compared with ANN-based methods.

## 4 RELATED WORK

From the perspective of the loss landscape, a good weight initialization is often located in a region with low loss values (Li et al., 2018). However, training deep neural networks is never an easy thing. After trails of layerwise training of deep nets (Hinton, 2007; Bengio et al., 2006), Glorot & Bengio (2010) provided a theoretical view on the prevention of exploding and vanishing gradients. They estimated the standard deviation of the weights from the number of layer neurons and initialized the biases to zero. He et al. (2015) extended their hypothesis to a piecewise linear case to cater to the growing prevalence of ReLU activation. Both methods are widely used in deep ANNs. There are also works for RNN. Saxe et al. (2014) examined the dynamics of learning and derived random

Table 2: Network performance comparison on the MNIST and CIFAR10 datasets with different initialization methods. A hyphen in the table indicates invalid training in a given setting.

| | Constant coding | Poisson coding | Retina-like coding | Event coding (N-MNIST) |
|---|---|---|---|---|
| MNIST ($k = 0.8,\ \lambda = 0.2,\ \theta = 1.0$) | | | | |
| Glorot's | 99.18 | 99.30 | 99.04 | 99.36 |
| He's | 99.18 | 99.26 | 99.14 | 99.35 |
| Wu's | 99.18 | 99.28 | **99.24** | 99.29 |
| Lee's | 99.18 | 99.27 | 99.14 | 99.33 |
| **Ours** | **99.27** | **99.35** | 99.20 | **99.39** |
| CIFAR10 ($k = 0.8,\ \lambda = 0.2,\ \theta = 1.0$) | Constant coding | Poisson coding | Retina-like coding | Event coding (CIFAR10-DVS) |
| Glorot's | 81.59 | 76.69 | 73.75 | - |
| He's | 82.95 | 81.86 | 76.36 | 66.9 |
| Wu's | 83.90 | 82.23 | 78.84 | 67.7 |
| Lee's | 83.05 | 81.03 | 73.07 | - |
| **Ours** | **87.62** | **83.06** | **82.98** | **68.2** |

orthogonal initial conditions for deep nonlinear networks. Le et al. (2015) used an identity matrix to initialize the RNN weight matrix and thus maintain the gradients.

Recent SNN studies have also included some initialization methods. For example, one can regard ANN-SNN conversion as an initialization method (Rathi et al., 2020). In addition, Lee et al. (2016) set the weights and thresholds using a manually tuned hyperparameter. They also proposed a complex BP error normalization procedure. Wu et al. (2018) suggested normalizing weights sampled from a standard uniform distribution, with no further explanation. Lee et al. (2020b) followed the recommendation of He et al. (2015) with a manually selected constant $\kappa$.

With the similar motivation of this work, many recent studies have focused on regularizing the spiking response. Zheng et al. (2021) and Kim & Panda (2020) proposed batch normalization along channel axes and time axes, respectively, for BPTT; both of which are layerwise normalization methods. In addition, Kim & Panda (2021) utilized a global unsupervised firing rate normalization. The above methods lack direct attention to neuronal dynamics. Sengupta et al. (2019), however, used dynamics of simple IF neuron proposing Spike-Norm for ANN-SNN conversion. In contrast, our work aims to improve the training for general first-order integrate-and-fire neurons by analyzing neuronal dynamics.

## 5 EXPERIMENTAL VALIDATION

### 5.1 EXPERIMENTAL SETTING AND DATASETS

To verify the training acceleration performance and the model performance against various other initialization methods, we conducted experiments on classification tasks involving different codings for the MNIST dataset (Lecun et al., 1998) and the CIFAR10 dataset (Krizhevsky et al., 2009). Before being fed into an SNN, images should be converted into spike trains. We implemented constant coding, Poisson coding, retina-like coding, and event coding for both datasets. Poisson coding and constant coding are popular rate coding schemes for ANN-SNN conversion (Rueckauer et al., 2017; Sengupta et al., 2019). Retina-like coding mimics the integration process of a bipolar cell (Zhu et al., 2019). The N-MNIST dataset was produced by converting MNIST into spike events with dimensions of size $34 \times 34$ using saccades (Orchard et al., 2015). The CIFAR10-DVS dataset comprises 10000 $128 \times 128$ event streams converted from CIFAR10 using a dynamic vision sensor (DVS) (Li et al., 2017).

Previous work has revealed that the accuracy of an SNN is sensitive to the choice of hyperparameters such as the neuron parameters (Fang et al., 2020). Therefore, we fixed the neuron parameters for a given dataset, as shown in Table 2, and compared our method with four other prevalent choices

(Glorot & Bengio (2010), He et al. (2015), Lee et al. (2016) and Wu et al. (2018)). An analysis of the hyperparameter sensitivity is presented in Sec. 6. For MNIST, we trained a 5-layer SNN. For the CIFAR10 dataset, we trained a VGG-9 model. For the CIFAR10-DVS dataset, we added one pooling layer and a fully-connected layer into the VGG-9 model. Time step in forwarding process is set to a constant value of 20. All the models were optimized using the Adam optimizer. The detailed training hyperparameters are shown in the appendix. Additionally, we conducted experiments for larger datasets, i.e., CIFAR100 and TinyImageNet, to compare to Batch Normalization (BN). See Appendix for the detailed descriptions and comments on experiments on larger datasets.

## 5.2 RESULTS

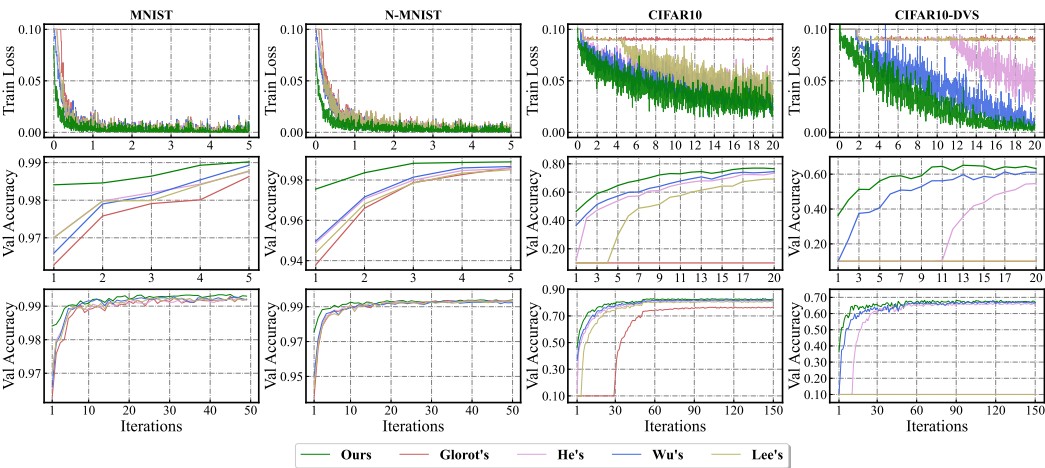

Figure 4: Training loss and validation accuracy curves on the MNIST, N-MNIST, CIFAR10, and CIFAR10-DVS datasets. The MNIST and CIFAR10 datasets are encoded with the Poisson distribution. The first two rows show the loss and accuracy curves in the earliest 5 and 20 epochs. The third row shows the validation accuracy curves throughout the whole training process.

In general, Fig. 4 and Table 2 show that good parameter initialization can not only accelerate training but also lead to some performance improvement. On MNIST, the different initialization methods achieve similar performance in validation accuracy with all four encoding schemes listed in Table 2, and our method performs slightly better for three of them. On CIFAR10, our method achieves performance improvement, showing accuracy increases of 4.67%, 1.20%, 6.62%, and 0.5% compared with He's method. As seen from the first row of Fig. 4, our method enables the model to generate early gradients in fewer iterations and cause the loss to drop faster. In more detail, our method allows the model to obtain a proper parameter distribution, which helps every layer coordinate with the input and output to better adapt to SNN training. This advantage enables network training to begin quickly, within a few iterations. In comparison, the other methods are slower than ours.

For MNIST, our method reaches 98% validation accuracy after only 1 epoch, while other methods reach this accuracy after about 2 or 3 epochs of training. For N-MNIST, our method reaches 98% accuracy after 2 epochs, while the others reach after 3 epochs. For CIFAR10, our method first achieves 60% accuracy at approximately the 4th epoch, whereas Wu's, He's, Lee's, and Glorot's methods reach this level later, respectively. For CIFAR10-DVS, our method and those of Wu and He reach 60% accuracy in order, while training fails under Lee's and Glorot's methods.

Due to the simplicity of MNIST, networks taking input encoded using each of the four schemes perform similarly. For CIFAR10, a model trained with constant coding achieves the highest accuracy because this data format is almost the same as the original data. Poisson coding performs worse than constant coding due to its more complicated coding rules. Retina-like coding leads to considerable degradation, probably due to the loss of some texture information. The results show that this coding scheme influences the other four initialization methods more than it does ours. The event data (CIFAR10-DVS) are sparse and complicated, and the amount of data is much smaller than that in CIFAR10, which increases the training difficulty and leads to performance degradation.

## 6  ANALYSIS OF EMPIRICAL RESULTS

To study whether the initialization strategy is sensitive to the hyperparameters of the model and the training process, we conducted many experiments. In these experiments, we used the MNIST dataset and the same 5-layer network as in Sec. 5. Training was performed for 3 epochs with a minibatch size of 64.

### 6.1  PERFORMANCE ANALYSIS ON DIFFERENT NEURONS

We performed an analysis of different neuron types, that is, different neuron parameter settings. While $\theta$ was fixed to 1.0, $k$ was sampled from 0.1 to 1.0, and $\lambda$ was sampled from 0.1 to 4.0. Fig. A8 compares the training speeds in this grid of experiments. For most $(k, \lambda)$ settings, our method shows obvious acceleration relative to the other methods. In particular, our method can be used to effectively train networks for which training cannot be successfully started using other initialization methods.

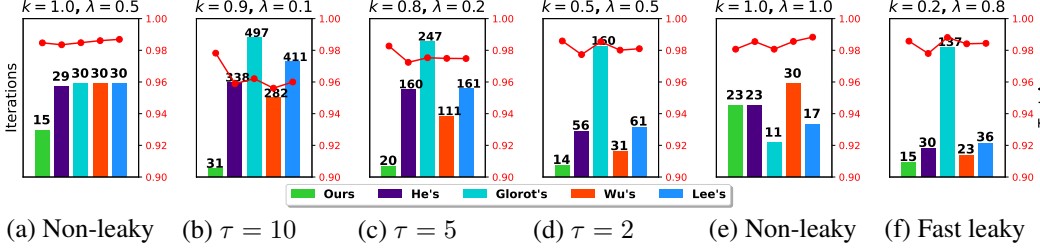

(a) Non-leaky   (b) $\tau = 10$   (c) $\tau = 5$   (d) $\tau = 2$   (e) Non-leaky   (f) Fast leaky

Figure 5: Validation accuracy (red polylines) and training iterations (colored bar charts, where the value indicates the number of iterations at which the accuracy first reaches the expected level of 60%) for first-order integrate-and-fire neurons with different $(k, \lambda)$ settings and initialization methods. Note that $\theta$ is set to 1.0.

We choose six representative groups to better illustrate this as shown in Fig. 5. For leaky neurons with different $\tau$ values, with decreasing $\tau$, the training speed and accuracy of all initialization methods improve. In this case, our method maintains high accuracy and achieves a training acceleration of more than $11\times$ compared to the baseline Glorot initialization. For nonleaky neurons, as shown in panels (a) and (e), the difference among methods is not obvious. As $\lambda$ decreases, our method enables faster training. For neurons that leak quickly ($k = 0.2$), Glorot's method performs poorly in terms of the start-up training speed but achieves comparable accuracy to our method. Among the faster methods, our method achieves the best accuracy.

Besides, instead of fixing $\theta$, we plot the heatmap in Fig. A9 after sampling $\theta$ from $\{0.5, 1, 2, 4\}$. This figure illustrates that our method is suitable for almost every chosen parameter and can help address the stalling of training caused by extremely large threshold due to the compensation of the initial biases.

### 6.2  PERFORMANCE ANALYSIS WITH DIFFERENT OPTIMIZERS AND SURROGATE FUNCTIONS

We selected $\tau = 5$ and 2 to test the sensitivity to the training hyperparameters. Two main factors that influence the gradient update are the optimizer and the surrogate function. Accordingly, Fig. 6 shows the training curves with different initialization methods, and the accuracy is marked in the figure in the order of the legends. Generally, if training can be successfully started via the gradient update with a given hyperparameter, the difference in accuracy is not huge. However, we can still observe an accuracy advantage with our method. For example, for $(0.8, 0.2, 1.0)$, the accuracy of Wu's and Lee's methods is 1.8% lower than ours. In addition, roughly, the initialization methods can be roughly arranged in the following order by the training speeds observed from the experiments: ours > Wu's > He's ≥ Lee's > Glorot's. Moreover, better training is observed in experiments with a smaller $\tau$.

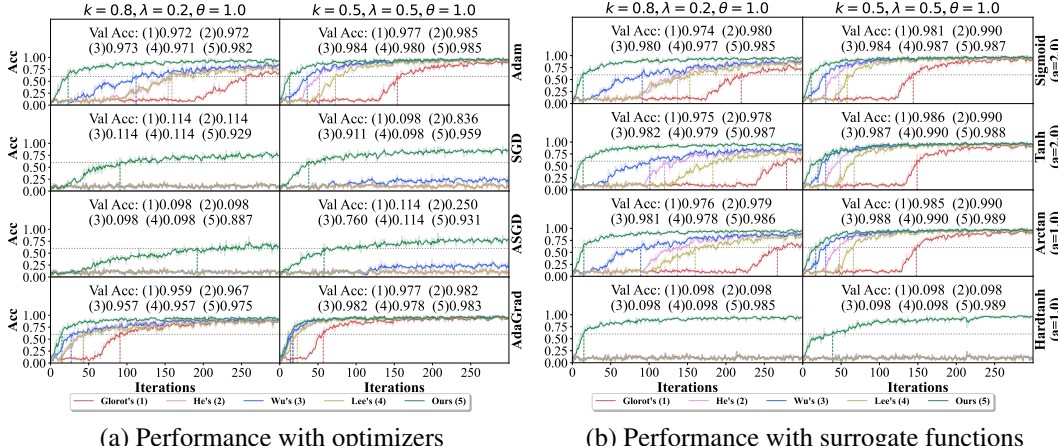

(a) Performance with optimizers  (b) Performance with surrogate functions

Figure 6: Training accuracy and validation accuracy using different training hyperparameters, different $(k, \lambda, \theta)$ settings and different initialization methods. The performance with four different optimizers (Adam, SGD, ASGD, AdaGrad) is shown in (a). The performance with four different surrogate functions (sigmoid, tanh, arctan, hardtanh) is presented in (b).

Regarding the optimizers, SGD(lr=0.5) and ASGD(lr=2.0) fail to activate fast early training for other initialization methods, yet our methods still works. Compared with Adam(lr=1e-3,$\beta$ =(0.9,0.99)), AdaGrad(lr=1e-2) seems less sensitive to the choice of the initialization methods.

With different surrogate functions, the order of the methods is maintained. Hardtanh produces coarse surrogate gradients and suppresses training under the other four methods. Surrogate functions that produce more fine-grained gradients all successfully activate training. Sigmoid(a=2.0) has some advantages in terms of training speed, but it is slightly inferior to tanh(a=2.0) and arctanh(a=1.0) in training accuracy.

## 7  DISCUSSIONS AND LIMITATIONS

The large number of experiments we conducted allows us to provide suggestions for SNN training. A good combination of hyperparameters is essential. Our initialization method can speed up training in most cases and adapt to more difficult neural parameters. In addition, we recommend the use of adaptive optimizers, as it helps accumulate the gradients. Finally, the arctan and tanh surrogate functions are suggested because they offer high accuracy and relatively high training speeds.

Apart from the advantages, our method still has some limitations in terms of neuronal dynamics. Although analyzing the response of spiking neurons and asymptotes is feasible in most cases, the theoretical results will be hampered if the input current carries more noise. The noise will have at least two impacts (Gerstner et al., 2014). One is that the neurons will fire even when the mean input is below zero. The other is that noise in the input current will cause the membrane potential to drift away from the theoretical asymptote. Furthermore, the noise will increase when we extend to wider and deeper networks. Therefore, for large datasets such as ImageNet, the asymptote should take the current noise into account.

## 8  CONCLUSION

In this work, we aim to enhance the efficient BPTT training of SNNs by proposing an initialization method that is consistent with the response of spiking neurons in initial training. Our method bridges the spiking neuron response with the wisdom of traditional deep learning training, which may have an influence on future research on topics such as ANN-SNN conversion with LIF neurons or other SNN training methods. Our experimental results validate that our initialization method shows superiority, versatility, and robustness over a vast training hyperparameter space.

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

# A APPENDIX

## A.1 APPENDIX FOR THEORETICAL ANALYSIS

Here, we provide theoretical proofs of the theorems in Sec. 2. These are the foundation of the analyses in this paper. Additionally, Fig. A7 is presented here to illustrate the surrogate functions and symmetric gradients.

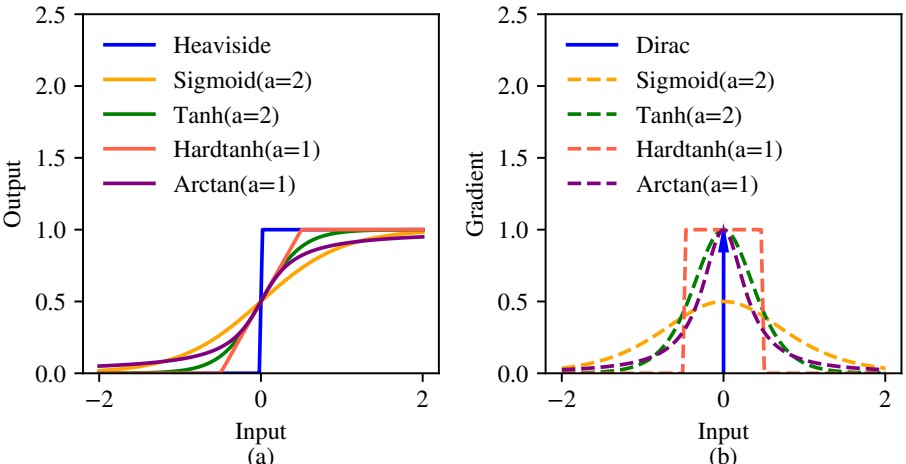

Figure A7: Surrogate functions and corresponding gradients.

Table A3: Surrogate functions for the non-differentiable model ($a$ is the parameter).

| **Surrogate function** $h$ | $h(x, a)$ | $h'(x, a)$ |
| --- | --- | --- |
| Sigmoid-like | $1/(1 + e^{-ax})$ | $a \cdot h(x, a)(1 - h(x, a))$ |
| Tanh-like | $\frac{1}{2}\tanh(ax) + \frac{1}{2}$ | $\frac{1}{2}a \cdot (1 - \tanh(ax)^2)$ |
| Hardtanh-like | $\text{clip}(ax + \frac{1}{2}, 0, 1)$ | $a \cdot \text{Heaviside}(\frac{1}{2a} - |x|)$ |
| Arctan-like | $\frac{1}{\pi}\arctan(\pi ax) + \frac{1}{2}$ | $a/(1 + (\pi ax)^2)$ |

**Theorem 1.** *For a first-order integrate-and-fire neuron as expressed in Eq. 2 ($u^t = kv^t + \lambda i^t, k \in (0, 1]$), the model takes a constant current $i^t = i$ as input and produces spikes as output. Let the discrete output firing rate w.r.t. $i$ be expressed as $f = \frac{T}{N}$, where $T$ is the total inference time and $N$ is the spike count. Then, the input-output response curve ($f$-$i$ curve) is estimated as follows:*

$$
\begin{aligned}
&\text{If } k = 1: \\
&\quad f = \begin{cases} 0, & i \le 0, \\ \frac{\lambda}{\theta}i, & 0 < i < \frac{\theta}{\lambda}, \\ 1, & i \ge \frac{\theta}{\lambda}. \end{cases} \\
&\text{If } k \in (0, 1): \\
&\quad f = \begin{cases} 0, & i \le 0, \\ (k-1)/\ln(1 + \frac{(k-1)\theta}{\lambda i}), & 0 < i < \frac{\theta}{\lambda} \cdot \frac{k-1}{e^{k-1}-1}, \\ 1, & i \ge \frac{\theta}{\lambda} \cdot \frac{k-1}{e^{k-1}-1}. \end{cases}
\end{aligned}
\tag{A1}
$$

*Proof.* From Eq. 2, the iterative functions are first converted into differential equations. Here, we use the scalar variable form because we are interested in the time interval between the resting state and the firing state for a constant input. Our goal is to determine the time $t^*$ of firing ($u(t^*) = \theta$)

with $u(0) = 0$. The discrete output firing rate $f = \frac{T}{N}$ can then be approximated as the inverse of the spiking interval $t^*$.

$$\frac{du}{dt} = (k-1)u + \lambda i \tag{A2}$$

One can use integration to obtain the expression for $u$ given a constant input. When $k = 1$, the integral is merely an accumulation of the scaled input. Considering $u(0) = 0$, we have the following expression:

$$u(t) = \lambda i \cdot t. \tag{A3}$$

Then, with $u(t^*) = \theta$, it is calculated that $t^* = \frac{\theta}{\lambda i}$. Thus, when $k = 1$,

$$f = \frac{T}{N} \approx \frac{1}{t^*} = \frac{\lambda}{\theta} i. \tag{A4}$$

For $i \geq \frac{\theta}{\lambda}$, the neuron constantly fires, and $f$ reaches its upper bound of 1. When $i \leq 0$, the neuron accumulates negative input and produce zero spikes, $f = 0$. When $k \in (0,1)$, Eq. A2 is an ordinary differential equation. By separating the integration variables $u$ and $t$, we obtain:

$$\frac{du}{(k-1)u + \lambda i} = dt.$$

Integrating both sides of the equation yields

$$\ln((1-k)u - \lambda i) = (k-1)t + C$$
$$(1-k)u - \lambda i = C'e^{(k-1)t},$$

where $C$ and $C'$ are the constants of integration. Considering $u(0) = 0$, we have $C' = -\lambda i$, and thus,

$$u(t) = \frac{\lambda i}{1-k}(1 - e^{-(1-k)t}). \tag{A5}$$

Then, by solving $u(t^*) = \theta$, we obtain $t^* = \frac{1}{k-1} \cdot \ln(1 + \frac{(k-1)\theta}{\lambda i})$. Hence,

$$f = \frac{T}{N} \approx \frac{1}{t^*} = (k-1)/\ln(1 + \frac{(k-1)\theta}{\lambda i}). \tag{A6}$$

When $i = \frac{\theta}{\lambda}\frac{k-1}{e^{k-1}-1}$, $f = 1$. Thus, when $i > \frac{\theta}{\lambda}\frac{k-1}{e^{k-1}-1}$, $f$ is clipped to 1. $\qquad \square$

**Theorem 2.** *For a first-order integrate-and-fire neuron as expressed in Eq. 2 ($u^t = kv^t + \lambda i^t, k \in (0,1]$), the slant asymptote of the $f$-$i$ curve satisfying $f \in (0,1)$ exists and can be uniformly expressed as follows:*

$$f_a = \frac{\lambda}{\theta}i + \frac{k-1}{2}. \tag{A7}$$

*Proof.* From Theorem 2, the relationship between $i$ and $f$ satisfying $f \in (0,1)$ is given as follows:

$$f = \begin{cases} \frac{\lambda}{\theta}i, & k = 1, \\ (k-1)/\ln(1 + \frac{(k-1)\theta}{\lambda i}), & k \in (0,1). \end{cases} \tag{A8}$$

Suppose that the asymptote is formulated as $f_a = K \cdot i + B$. Then, for $k = 1$, $K = \frac{\lambda}{\theta}$, and $B = 0$. For $k \in (0,1)$, if a slant asymptote exists, then $\lim_{i\to\infty}\frac{f}{i} \to K$ and $\lim_{i\to\infty}[f - Ki] \to B$.

$$\lim_{i \to \infty} \frac{f}{i} = \lim_{i \to \infty} \frac{k-1}{i \ln(1 + \frac{\theta(k-1)}{\lambda i})}$$

$$= \lim_{i \to \infty} \frac{\frac{k-1}{i}}{\ln(\lambda i + \theta(k-1)) - \ln(\lambda i)}$$

$$= \lim_{i \to \infty} \frac{-\frac{k-1}{i^2}}{\frac{\lambda}{\lambda i + \theta(k-1)} - \frac{\lambda}{\lambda i}} \quad \text{(L'Hospital's Rule)}$$

$$= \lim_{i \to \infty} \frac{\lambda i + \theta(k-1)}{\theta i}$$

$$= \frac{\lambda}{\theta} \quad \text{(L'Hospital's Rule)}$$

Thus, a slant asymptote exists for $k \in (0,1)$, and $K = \frac{\lambda}{\theta}$ for all $k \in (0,1]$. For $k \in (0,1)$,

$$B = \lim_{i \to \infty} \left[ \frac{k-1}{\ln(1 + \frac{\theta(k-1)}{\lambda i})} - \frac{\lambda}{\theta} i \right]$$

$$= \lim_{i \to \infty} \frac{k - 1 - \frac{\lambda}{\theta} i [\ln(\lambda i + \theta(k-1)) - \ln(\lambda i)]}{\ln(\lambda i + \theta(k-1)) - \ln(\lambda i)}$$

$$= \lim_{i \to \infty} \frac{-\frac{\lambda}{\theta} \ln(\lambda i + \theta(k-1)) - \frac{\lambda}{\theta} \frac{\lambda i}{\lambda i + \theta(k-1)} + \frac{\lambda}{\theta} \ln(\lambda i) + \frac{\lambda}{\theta}}{\frac{\lambda}{\lambda i + \theta(k-1)} - \frac{\lambda}{\lambda i}} \quad \text{(L'Hospital's Rule)}$$

$$= \lim_{i \to \infty} \frac{\lambda}{\theta} \frac{\ln(\lambda i) - \ln(\lambda i + \theta(k-1)) + 1 - \frac{\lambda i}{\lambda i + \theta(k-1)}}{\frac{\lambda}{\lambda i + \theta(k-1)} - \frac{\lambda}{\lambda i}}$$

$$= \lim_{i \to \infty} \frac{\lambda}{\theta} \frac{\frac{1}{i} - \frac{\lambda(\lambda i + \theta(k-1)) + \lambda \theta(k-1)}{(\lambda i + \theta(k-1))^2}}{-\frac{\lambda^2}{(\lambda i + \theta(k-1))^2} + \frac{1}{i^2}} \quad \text{(L'Hospital's Rule)}$$

$$= \lim_{i \to \infty} \frac{\lambda}{\theta} \frac{\theta^2(k-1)^2 i}{2\lambda \theta(k-1)i + \theta^2(k-1)^2}$$

$$= \frac{\lambda}{\theta} \frac{\theta(k-1)}{2\lambda} \quad \text{(L'Hospital's Rule)}$$

$$= \frac{k-1}{2}$$

When $k = 1$, $B = 0$. Thus, $f_a = \frac{\lambda}{\theta} i + \frac{k-1}{2}$ is the uniform slant asymptote for all $k \in (0,1]$.

$\square$

## A.2 APPENDIX FOR EXPERIMENTAL RESULTS

The detailed hyperparameters are presented in Table A4. Note that the data in CIFAR10-DVS and N-MNIST take the form of address event representation (AER). To split the events into T slices that match the input streams of our network, we utilized the technique referenced in Appendix F of Fang et al. (2020) to preprocess the data.

**Performance Analysis under Different Neuron Configurations**

Eq. 2 shows that the neuron types and firing patterns are controlled by $k$ and $\lambda$, which may vary in the biological brain. To cover as many of the neuron models used in the existing neuroscience literature as possible, we discretely sampled $k$ from 0.1 to 1.0 and $\lambda$ from 0.1 to 4.0. In each experiment, we compared four other methods with our initialization method in two aspects: training speed and accuracy. The four methods considered for comparison were proposed by Glorot & Bengio (2010), He et al. (2015), Lee et al. (2016) and Wu et al. (2018).

Table A4: Detailed hyperparameter settings in the experimental validation. The networks were designed to fit the different shapes of input data.

| | MNIST | N-MNIST | CIFAR10 | CIFAR10-DVS |
|---|---|---|---|---|
| Input size | $28 \times 28$ | T$\times 32 \times 32$ | $32 \times 32$ | T$\times 128 \times 128$ |
| Timestep | 20 | 20 | 20 | 20 |
| Network | conv $3 \times 3 \times 32$
LIF neuron
pool $2 \times 2$
conv $3 \times 3 \times 32$
LIF neuron
pool $2 \times 2$
fc 500
LIF neuron
fc 100
LIF neuron
fc 10 | conv $3 \times 3 \times 32$
LIF neuron
pool $2 \times 2$
conv $3 \times 3 \times 32$
LIF neuron
pool $2 \times 2$
fc 500
LIF neuron
fc 100
LIF neuron
fc 10 | conv $3 \times 3 \times 128$
LIF neuron
conv $3 \times 3 \times 128$
LIF neuron
pool $2 \times 2$
conv $3 \times 3 \times 256$
LIF neuron
conv $3 \times 3 \times 256$
LIF neuron
pool $2 \times 2$
conv $3 \times 3 \times 256$
LIF neuron
conv $3 \times 3 \times 256$
LIF neuron
conv $3 \times 3 \times 256$
LIF neuron
pool $2 \times 2$
fc 1024
LIF neuron
dropout
fc 10 | conv $3 \times 3 \times 64$
LIF neuron
conv $3 \times 3 \times 128$
LIF neuron
pool $2 \times 2$
conv $3 \times 3 \times 128$
LIF neuron
conv $3 \times 3 \times 128$
LIF neuron
pool $2 \times 2$
conv $3 \times 3 \times 128$
LIF neuron
pool $2 \times 2$
conv $3 \times 3 \times 128$
LIF neuron
pool $2 \times 2$
conv $3 \times 3 \times 128$
LIF neuron
pool $2 \times 2$
fc 1024
LIF neuron
dropout
fc 1024
LIF neuron
dropout
fc 10 |
| Epoch | 50 | 50 | 150 | 150 |
| Batch size | 32 | 32 | 32 | 16 |
| LR | 0.001 | 0.001 | 0.001 | 0.001 |
| Optimizer | Adam(0.9,0.99) | Adam(0.9,0.99) | Adam(0.9,0.99) | Adam(0.9,0.99) |
| Scheduler | Cosine(T$_{max}$=50) | Cosine(T$_{max}$=50) | Multistep
(T=[50,90,130]) | Multistep
(T=[50,90,130]) |
| Surrogate | Arctan(a=2.0) | Arctan(a=2.0) | Arctan(a=2.0) | ArcTan(a=2.0) |

The exact iteration step at which the training accuracy first reached a certain expected level (60%) was recorded to represent the training speed. In Fig. A8, green bars represent our method, while bars in other colors represent other methods. It is intuitive that our method reaches the expected accuracy faster or at least at the same performance level for most neuron types compared to other methods; this is more obvious when $\lambda$ falls in the range of $(-\infty, 0.6)$ or $(2.0, +\infty)$. In particular, there are no valid records at all using other methods in some cases in which $\lambda$ is set to 0.1, 0.2, 8.0, and so on. However, even in these cases, the training accuracy of our method still reaches the expected level in a very short time.

In addition to our method's superiority in terms of training speed, we measured its performance in terms of the validation accuracy to check its robustness on classification tasks under various configurations. In Fig. A9, the classification accuracies of five initialization methods under different neuron configurations are shown as heatmaps. Our method can be used to effectively and efficiently

Table A5: Optimizers and their parameters used to test the versatility.

| Optimizer | Learning rate | Other Parameters |
|---|---|---|
| Adam | 0.001 | betas=(0.9, 0.99) |
| SGD | 0.5 | momentum=0.9 |
| ASGD | 2.0 | lambda=0.000001, alpha=0.75 |
| AdaGrad | 0.01 | - |

train SNNs with almost all neurons types, while other methods fail to train networks with some neuron types.

To further analyze whether the proposed method exhibits versatility for various choices of optimizers and surrogated funtions, eight representative groups of $(k, \lambda, \theta)$ were selected: (0.3, 0.1, 1.0), (0.8, 0.2, 1.0), (0.8, 0.5, 1.0), (0.2, 0.8, 1.0), (1.0, 1.0, 1.0), (0.5, 4.0, 1.0), (0.6, 0.3, 4.0) and (0.4, 0.4, 4.0).

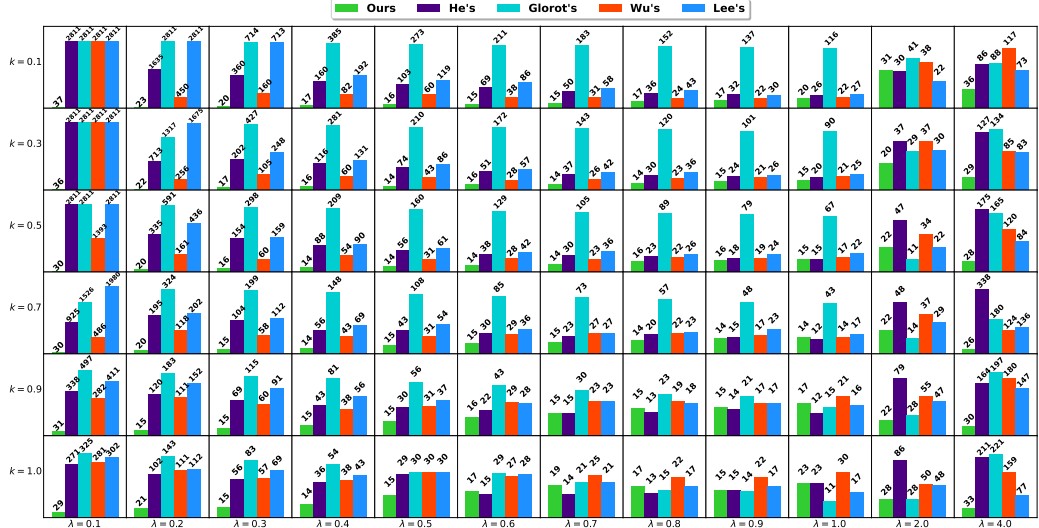

Figure A8: Training iterations at which the accuracy first reaches the expected level (60%) for first-order integrate-and-fire neurons with different $(k, \lambda)$ settings and initialization methods. Note that $\theta$ is set to 1.0. The total number of iterations of 3 training epochs is 2811. A value of 2811 in this figure means that in the corresponding setting, the network cannot be trained to reach the expected training accuracy level.

**Versatility Across Optimizers**

During model training, different optimizers may be used to minimize the loss function. To test the versatility of our method across various optimizers, we chose four classic and prevalent optimizers for model training, i.e., Adam, SGD, ASGD, and AdaGrad. The detailed optimizer parameters are listed in Table A5.

We present the accuracy of training in Table A7. In Fig. A10, we show the training curves during the first 500 iterations to illustrate the results in more detail. As shown in this figure, when SGD or ASGD is chosen to optimize the network, our method demonstrates an advantage in accelerating the training in the start-up phase with the accuracy reaching a good level within 100 iterations in most cases. For example, when $k = 0.8$, $\lambda = 0.5$, and $\theta = 1.0$, Wu's and He's methods allow training to begin early but with a slower speed, achieving an accuracy of approximately 90% after 500 iterations, while Glorot's and Lee's methods show much worse performance. When Adam or AdaGrad is chosen to optimize the network, despite the similar performance of all methods in some cases (for example the 3rd-5th rows in the figure), our method outperforms the others in accelerating

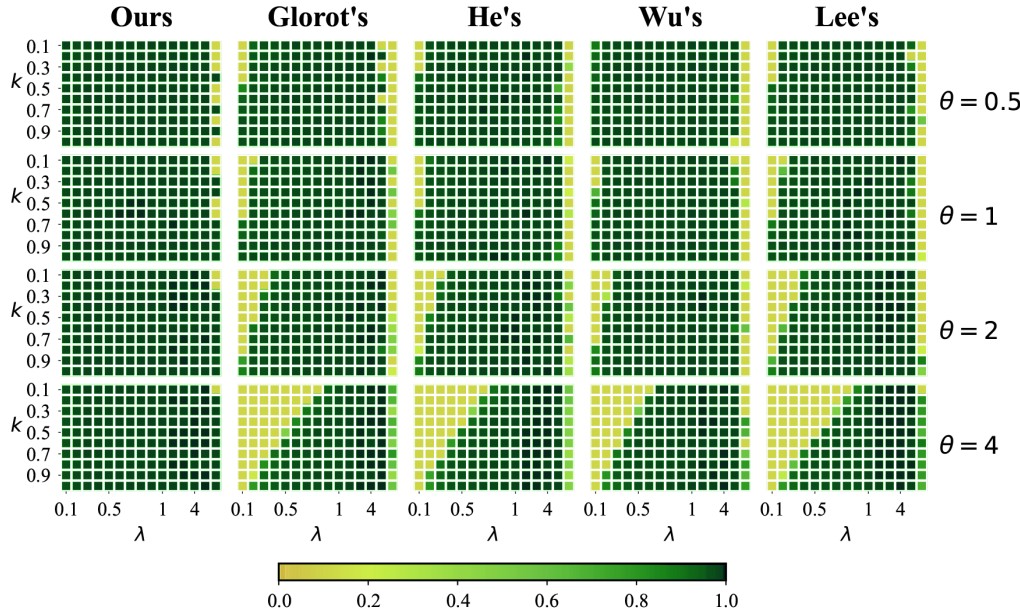

Figure A9: Training accuracy heatmaps for first-order integrate-and-fire neurons with different $(k, \lambda, \theta)$ settings and initialization methods. The color represents the validation accuracy in the given setting.

the training process. In addition, our weight initialization for SNNs enables valid training in some difficult cases in which training totally fails with other initialization methods, such as $(k = 0.4, \lambda = 0.4, \theta = 0.4)$.

**Versatility Across Surrogate Functions**

Various surrogate functions may be chosen for SNN training w.r.t. different needs and tasks. As listed in Table A3, four surrogate functions (sigmoid, tanh, hardtanh, and arctan) are popularly used in BPTT. As described in Sec. 2.1, the input is subjected to the Heaviside function to perform the forward pass, while the backward gradient is computed using surrogate functions. We trained the 5-layer SNN using surrogate functions consistent with those in Table A3 by adjusting the parameter $a$.

The validation accuracy after 3 epochs of training is presented in Table A7, and the training curves from the first 220 iterations with the four surrogate functions are shown in Fig. A11. The curves in green show our method's effectiveness. Our method reaches acceptable accuracy within relativelt few iterations (fewer than 50 in most cases), while in comparison, the decline in the network loss starts more slowly under other initialization methods, taking parameter sets of (0.8, 0.2, 1.0), (0.8, 0.5, 1.0), (0.2, 0.8, 1.0) and (0.5, 4.0, 1.0) as examples. Especially when $k$ and $\lambda$ have small values or $\theta$ has a large value, other initialization methods cannot generate effective gradients for BP, leading to training failure. For almost all sets of surrogate functions and neuronal parameters, our method enables neuron firing and early generation of network gradients. Fig. A11 reveals that our method enables models to start training faster and to end training with considerably greater classification accuracy.

## A.3    COMPARISON TO BATCH NORMALIZATION (BN)

As Eq. 11 implies, the variances of input current from the adjacent layers have been scale to be identical when using our initialization method, which can be viewed as a kind of normalization of current before the start of training. The difference lies in the term of validity. Normalization methods such as Kim & Panda (2020), Kim & Panda (2021) are normalizing current throughout training. To clarify this, we examined the validation accuracy with or without normalization methods on CIFAR100 and TinyImageNet datasets. We chose vanilla BN (Ioffe & Szegedy, 2015) and BNTT

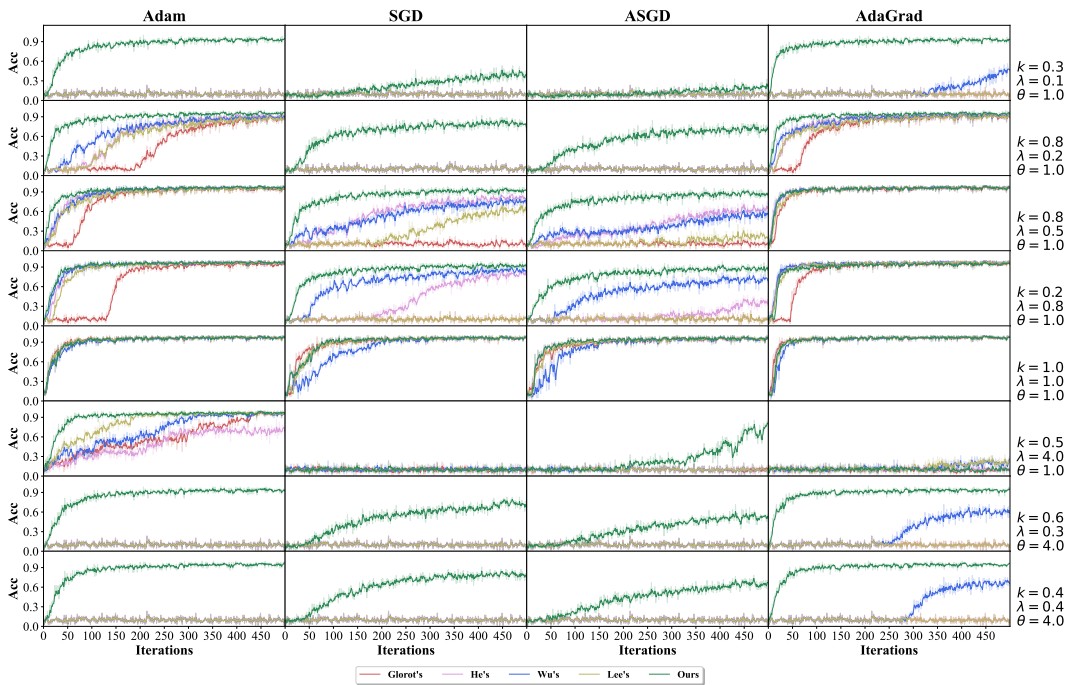

Figure A10: Training accuracy curves obtained using four different optimizers, different $(k, \lambda, \theta)$ settings and different initialization methods.

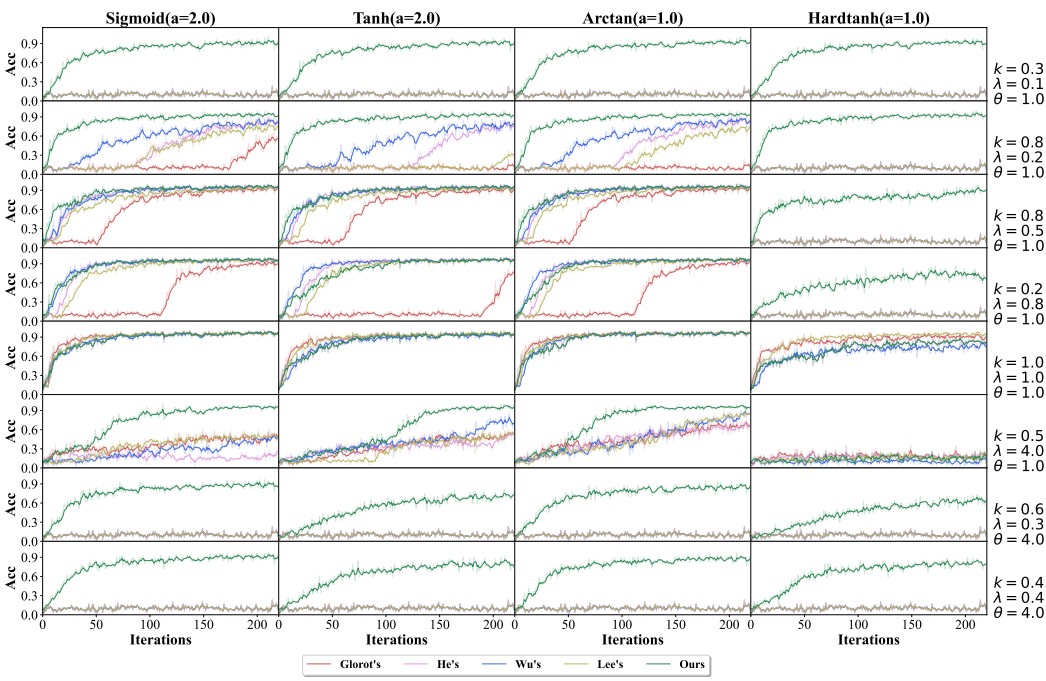

Figure A11: Training curves obtained using four different surrogate functions, different $(k, \lambda, \theta)$ settings and different initialization methods.

(batch normalization through time) (Kim & Panda, 2020) for comparison. When the network is without BN, a dropout is implemented in place of BN. $(k = 0.99, \lambda = 1.0)$ is the default setting inKim & Panda (2020). The validation accuracy curve is plotted in Fig. A12. The use of BNTT make the effect of initialization insignificant as it works throughout training. Table A8 shows that with BN

Table A6: Validation accuracy achieved when training on MNIST for 3 epochs using four different optimizers, different $(k, \lambda, \theta)$ settings and different initialization methods. A hyphen in this table indicates invalid training in a given setting.

| | $k = 0.3,\ \lambda = 0.1,\ \theta = 1.0$ | | | | | $k = 0.8,\ \lambda = 0.2,\ \theta = 1.0$ | | | | |
| | **Ours** | Glorot's | He's | Wu's | Lee's | **Ours** | Glorot's | He's | Wu's | Lee's |
|---|---|---|---|---|---|---|---|---|---|---|
| Adam | 0.979 | - | - | - | - | 0.982 | 0.972 | 0.972 | 0.973 | 0.971 |
| SGD | 0.776 | - | - | - | - | 0.929 | - | - | - | - |
| ASGD | 0.636 | - | - | - | - | 0.887 | - | - | - | - |
| AdaGrad | 0.966 | 0.878 | 0.898 | 0.876 | 0.740 | 0.975 | 0.959 | 0.967 | 0.956 | 0.957 |
| | $k = 0.8,\ \lambda = 0.5,\ \theta = 1.0$ | | | | | $k = 0.2,\ \lambda = 0.8,\ \theta = 1.0$ | | | | |
| | **Ours** | Glorot's | He's | Wu's | Lee's | **Ours** | Glorot's | He's | Wu's | Lee's |
| Adam | 0.984 | 0.980 | 0.986 | 0.983 | 0.984 | 0.984 | 0.978 | 0.988 | 0.982 | 0.989 |
| SGD | 0.975 | 0.921 | 0.959 | 0.964 | 0.945 | 0.968 | - | 0.950 | 0.956 | 0.884 |
| ASGD | 0.958 | 0.165 | 0.910 | 0.918 | 0.866 | 0.951 | - | 0.905 | 0.926 | 0.612 |
| AdaGrad | 0.984 | 0.978 | 0.984 | 0.984 | 0.982 | 0.977 | 0.978 | 0.986 | 0.985 | 0.986 |
| | $k = 1.0,\ \lambda = 1.0,\ \theta = 1.0$ | | | | | $k = 0.5,\ \lambda = 4.0,\ \theta = 1.0$ | | | | |
| | **Ours** | Glorot's | He's | Wu's | Lee's | **Ours** | Glorot's | He's | Wu's | Lee's |
| Adam | 0.987 | 0.986 | 0.987 | 0.986 | 0.986 | 0.986 | 0.987 | 0.984 | 0.985 | 0.986 |
| SGD | 0.985 | 0.982 | 0.985 | 0.983 | 0.985 | - | - | - | - | - |
| ASGD | 0.986 | 0.981 | 0.986 | 0.981 | 0.984 | 0.981 | - | - | - | - |
| AdaGrad | 0.986 | 0.984 | 0.986 | 0.985 | 0.986 | 0.983 | - | - | 0.977 | 0.984 |
| | $k = 0.6,\ \lambda = 0.3,\ \theta = 4.0$ | | | | | $k = 0.4,\ \lambda = 0.4,\ \theta = 4.0$ | | | | |
| | **Ours** | Glorot's | He's | Wu's | Lee's | **Ours** | Glorot's | He's | Wu's | Lee's |
| Adam | 0.980 | - | - | - | - | 0.984 | - | - | - | - |
| SGD | 0.895 | - | - | - | - | 0.923 | - | - | - | - |
| ASGD | 0.841 | - | - | - | - | 0.876 | - | - | - | - |
| AdaGrad | 0.968 | 0.882 | 0.889 | 0.910 | 0.743 | 0.972 | 0.906 | 0.924 | 0.926 | 0.752 |

or BNTT, the gaps of the accuracy of different initialization methods are not obvious. But when the network is without BN, our method can also work under (0.5, 0.5) configuration. Therefore, we are not stating that our initialization method can outperform some well-designed learning algorithms. Instead, we believe our method is a useful supplement to make SNNs trainable on larger datasets.

Table A7: Validation accuracy achieved when training on MNIST for 3 epochs using four different surrogate functions, different $(k, \lambda, \theta)$ settings and different initialization methods. A hyphen in this table indicates invalid training in a given setting.

| | $k = 0.3,\ \lambda = 0.1,\ \theta = 1.0$ | | | | | $k = 0.8,\ \lambda = 0.2,\ \theta = 1.0$ | | | | |
| | **Ours** | Glorot's | He's | Wu's | Lee's | **Ours** | Glorot's | He's | Wu's | Lee's |
|---|---|---|---|---|---|---|---|---|---|---|
| Hardtanh($a$=1.0) | 0.982 | - | - | 0.310 | - | 0.985 | 0.974 | 0.980 | 0.980 | 0.977 |
| Arctan($a$=1.0) | 0.983 | - | - | - | - | 0.984 | 0.972 | 0.980 | 0.980 | 0.976 |
| Tanh($a$=2.0) | 0.984 | - | - | - | - | 0.986 | 0.976 | 0.979 | 0.981 | 0.978 |
| Sigmoid($a$=2.0) | 0.984 | - | - | - | - | 0.985 | - | - | - | - |

| | $k = 0.8,\ \lambda = 0.5,\ \theta = 1.0$ | | | | | $k = 0.2,\ \lambda = 0.8,\ \theta = 1.0$ | | | | |
| | **Ours** | Glorot's | He's | Wu's | Lee's | **Ours** | Glorot's | He's | Wu's | Lee's |
|---|---|---|---|---|---|---|---|---|---|---|
| Hardtanh($a$=1.0) | 0.987 | 0.985 | 0.987 | 0.984 | 0.986 | 0.987 | 0.983 | 0.990 | 0.986 | 0.988 |
| Arctan($a$=1.0) | 0.988 | 0.983 | 0.986 | 0.987 | 0.989 | 0.988 | 0.986 | 0.991 | 0.987 | 0.991 |
| Tanh($a$=2.0) | 0.989 | 0.987 | 0.987 | 0.988 | 0.990 | 0.988 | 0.984 | 0.990 | 0.989 | 0.991 |
| Sigmoid($a$=2.0) | 0.988 | - | - | - | - | 0.986 | - | - | - | - |

| | $k = 1.0,\ \lambda = 1.0,\ \theta = 1.0$ | | | | | $k = 0.5,\ \lambda = 4.0,\ \theta = 1.0$ | | | | |
| | **Ours** | Glorot's | He's | Wu's | Lee's | **Ours** | Glorot's | He's | Wu's | Lee's |
|---|---|---|---|---|---|---|---|---|---|---|
| Hardtanh($a$=1.0) | 0.988 | 0.988 | 0.988 | 0.984 | 0.989 | 0.985 | 0.987 | 0.980 | 0.984 | 0.986 |
| Arctan($a$=1.0) | 0.986 | 0.987 | 0.986 | 0.981 | 0.984 | 0.987 | 0.987 | 0.973 | 0.973 | 0.978 |
| Tanh($a$=2.0) | 0.988 | 0.988 | 0.988 | 0.990 | 0.987 | 0.988 | 0.983 | 0.985 | 0.984 | 0.980 |
| Sigmoid($a$=2.0) | 0.987 | 0.986 | 0.987 | 0.887 | 0.989 | 0.199 | 0.197 | - | - | 0.176 |

| | $k = 0.6,\ \lambda = 0.3,\ \theta = 4.0$ | | | | | $k = 0.4,\ \lambda = 0.4,\ \theta = 4.0$ | | | | |
| | **Ours** | Glorot's | He's | Wu's | Lee's | **Ours** | Glorot's | He's | Wu's | Lee's |
|---|---|---|---|---|---|---|---|---|---|---|
| Hardtanh($a$=1.0) | 0.979 | - | - | - | - | 0.983 | - | - | - | - |
| Arctan($a$=1.0) | 0.917 | - | - | - | - | 0.953 | - | - | - | - |
| Tanh($a$=2.0) | 0.968 | - | - | - | - | 0.976 | - | - | - | - |
| Sigmoid($a$=2.0) | 0.887 | - | - | - | - | 0.951 | - | - | - | - |

Table A8: Accuracy comparison with different normalization technique training with VGG11 on large datasets. A hyphen in this table indicates invalid training in a given setting.

| Dataset | $(k, \lambda)$ | Technique | Initialization method | | | | |
| | | | Glorot's | He's | Wu's | Lee's | Ours |
|---|---|---|---|---|---|---|---|
| CIFAR100 | $(0.5, 0.5)$ | w/o BN, Dropout | - | - | - | - | **54.94** |
| CIFAR100 | $(0.99, 1.0)$ | w/o BN, Dropout | - | 54.03 | **54.46** | - | 53.79 |
| CIFAR100 | $(0.99, 1.0)$ | w/ BN | 57.18 | **57.19** | 56.45 | 56.64 | 57.02 |
| CIFAR100 | $(0.99, 1.0)$ | w/ BNTT | **65.89** | 65.55 | 64.91 | 65.65 | 65.44 |
| TinyImageNet | $(0.5, 0.5)$ | w/o BN | - | - | - | - | **44.16** |
| TinyImageNet | $(0.99, 1.0)$ | w/ BN | 45.97 | 47.18 | **47.60** | 45.03 | 47.51 |
| TinyImageNet | $(0.99, 1.0)$ | w/ BNTT | 57.03 | 57.09 | 56.90 | **57.19** | 56.80 |

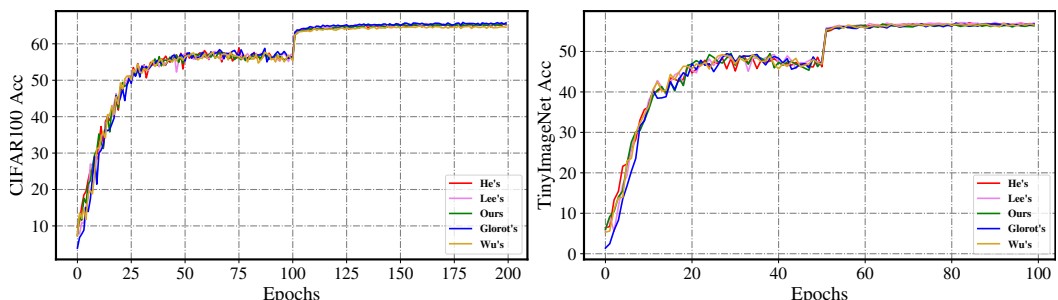

Figure A12: BNTT Training curves obtained using different initialization methods under the ($k$=0.99, $\lambda$=1.0) configuration.

