# OpenReview forum: "Accelerating Training of Deep Spiking Neural Networks with Parameter Initialization"
_ICLR.cc/2022/Conference — ICLR 2022 Submitted_

### Official Review · Reviewer_ZLvg · 2021-10-30

**Correctness:** 3
**Technical Novelty And Significance:** 3
**Empirical Novelty And Significance:** 3
**Recommendation:** 6
**Confidence:** 5

**Main Review:**

Strengths:
This paper proposes a weight initialization method to enhance the efficient BPTT training of SNNs. The derivation and validation of the weight initialization method appears to be sound. It is interesting to see the authors relate the spiking neuron response to the traditional deep learning training. This work may also inspire research like ANN-to-SNN conversions with LIF neurons.
Weaknesses:
(1) The related work section should be introduced in more detail. Is there any other study about weight initialization of SNN?
(2) Why is the weight initialization more effective on the CIFAR10 than the MNIST?
(3) I suggest to demonstrate the effect of the initialization method on more complex dataset..


**Summary Of The Paper:**

 This paper focuses on the parameter initialization problem of training SNNs. The authors derive the theoretical response of spiking neurons, and propose an initialization method based on slant asymptote, which can overcome the gradient vanishing problem. The results show that the proposed method can effectively improve training speed and accuracy.

**Summary Of The Review:**

Overall, it is a nice paper with solid theoretical basis. It has the potential to convince researchers to adopt SNNs and exploit learning of deep SNN. I consider to raise my score if the authors can resolve the concerns above.

---

> ### Author Response · Authors · 2021-11-18
> **Response to Reviewer ZLvg**
>
> We appreciate your constructive comments and suggestions. We are delighted that you find our work is with a solid theoretical basis. The response to your concerns are listed below:
>
> > The related work section should be introduced in more detail. Is there any other study about weight initialization of SNN?
>
> The field of SNN is now flourishing, and the neurons described in different works are usually not the same. As mentioned in the related work, "Recent SNN studies have also included some initialization methods". However, these initialization methods are usually proposed for the training of specific models without further analysis. There is no universal initialization method for SNNs for now, let alone initialization that can improve training. Our work aims to find a general initialization method to improve the spiking response by standardizing the neuron model and then to improve the SNN training.
> To improve spiking response, there are some works related to SNN training techniques. We have included them in our related work.
>
>
> > Why is the weight initialization more effective on the CIFAR10 than the MNIST?
>
> There are differences in the difficulty of classification between the two datasets. The features in MNIST are mostly edge, and there are fewer textures. Therefore, under the condition of sufficient network capacity, networks can learn excellent results. Initializing these SNNs will not have a great impact. In contrast, the CIFAR10 dataset contains natural images with rich textures. The choice of the inference time $T$ will make the learning more difficult. Therefore, the accuracy of CIFAR10 will be more affected by the initialization hyperparameters. Our experiments demonstrate superiority in most cases on the MNIST and CIFAR10 datasets.
>
> >  I suggest to demonstrate the effect of the initialization method on more complex dataset..
>
> Thank you for your suggestions. We have included the experiments on complex datasets (CIFAR100 and TinyImageNet) in the manuscript. Please refer to the general response 2 for more detail.

---

### Official Review · Reviewer_LCr2 · 2021-11-01

**Correctness:** 4
**Technical Novelty And Significance:** 3
**Empirical Novelty And Significance:** 3
**Recommendation:** 6
**Confidence:** 4

**Main Review:**

Whereas previous work on Deep SNN has often borrowed techniques from standard ANNs, this paper goes beyond that and develops SNN-specific methods that can accelerate Deep SNN training and lead to better accuracies. This is achieved from theoretical analysis of SNN neuron models and network dynamics, and the results seem to indicate that the method is generally superior to non-SNN-specific methods under a variety of parameters and datasets. If this method generalizes to larger tasks and networks, this could become a new standard for SNN training, and therefore a significant contribution to the SNN literature.

Strenghts:
+ derivation of the weight initialization method from theory
+ convincing results of experimental validation on a variety of datasets
+ comparison to a number of standard initialization methods from the ANN world,which do not have SNN-specific adaptations
+ possibly very broad application field, could become a new standard

Weaknesses:
- occasional unclear writing and grammar errors which should be fixed
- tested only on MNIST and CIFAR variants, it is unclear whether the method generalizes to larger networks

Recommendation:
- I would suggest including at least one larger experiment that shows that the weight initialization scales to larger networks, and is not an effect of the relatively easy datasets that were considered

**Summary Of The Paper:**

The paper investigates the effects of weight initialization for deep spiking neural networks (SNNs) on training speed and final accuracy. The main finding is that popular initialization methods for conventional ANNs and RNNs do not match the specific dynamics of SNNs, in particular they ignore the need to have sufficient firing early on to generate good gradients. The paper derives a theoretical first-order approximation on the response curve, and from it a weight initialization scheme that initializes the weights in a region where neurons fire from the first epoch on, but also avoid explosion or reduction of activities throughout the network. In various experiments on standard and spiking variants of MNIST and CIFAR, the initialization scheme shows good performance and fast convergence for different neuron types (e.g. non-leaking vs. fast leaking), and different encoding schemes. For CIFAR10 the results are significantly better than for other initialization schemes.

**Summary Of The Review:**

The presented method could become important for further studies on Deep SNNs, as a replacement of poorly fitting weight initialization schemes borrowed from ANNs. It remains to be shown that the method scales to more difficult tasks and larger networks. If the method still works there, this could be a paper with very high impact.

---

> ### Author Response · Authors · 2021-11-18
> **Response to Reviewer LCr2**
>
> Thank you for the positive feedback. We are encouraged that you find our presented method important and convincing. In the meantime, we have further improved our paper. We would like to address your concerns below.
>
> > Occasionally unclear writing and grammar errors which should be fixed
>
> Thanks for pointing it out! We thoroughly proofread and edited our manuscripts with the help of an English proofreading agency. Besides, we have fixed the unclear writing.
>
> > I would suggest including at least one larger experiment that shows that the weight initialization scales to larger networks, and is not an effect of the relatively easy datasets that were considered
>
> Thanks for your suggestions. We have included the experiments on complex datasets (CIFAR100 and TinyImageNet) in the manuscript. Please refer to the general response 2 for more details.

---

> > ### Comment · Reviewer_LCr2 · 2021-11-29
> > **Response to Author Comments**
> >
> > I´d like to thank the authors for improving the writing quality of the manuscript, and addressing the weakness of missing experiments on larger datasets. Unfortunately that analysis has shown that the method still has issues with larger and deeper networks, which is a serious concern about the general usefulness of the method. I do appreciate though the manner in which this has been addressed in the updated manuscript, both in terms of pointing out limitations, and adding experiments to investigate the problems. I have to lower my score overall due to the limitations of the method, but will keep it at 6, i.e. above acceptance threshold.

---

> > > ### Author Response · Authors · 2021-11-30
> > > **Reply to Reviewer LCr2**
> > >
> > > We would like to express our great appreciation for your reading and patience on our response, as well as your careful and objective comments on our work. We also sincerely accept your judgment after the review period.
> > >
> > > For in-depth exploration on training of spiking neural networks, we always have been maintaining an objective attitude and continuous passion, which we also keep when doing the experiments on larger datasets in this work. We believe our results and analysis would provide us with valuable reference and experience in our later research.
> > >
> > > For the results of our experiments on CIFAR100 and TinyImageNet datasets, we would like to give some supplementary interpretations. Overall, under some configurations, our method got fairly positive results, such as the results shown at row 1 and row 5 in Table A8 in the paper. At the same time, in other configurations, our results also basically reach the same level as other methods do. We would keep focusing on training efficiency on large-scale datasets in the future work.
> > >
> > > We consider the issues with larger and deeper networks for our method are important but not that urge. There are many factors that affect learning in the space of SNN training hyperparameters using large datasets. We know little about the impact of other factors on training for now(such as the impact of noise or learning algorithms); hence we believe our work has completed the theoretical discussion of SNN initialization methods at the moment. That is why the detailed discussion of limitations is considered as future work, which will not reduce the value of this paper.
> > >
> > > Thanks again for your reply. We would always dedicate ourself studying efficient training for spiking neural networks.

---

### Official Review · Reviewer_xJ5p · 2021-11-03

**Correctness:** 3
**Technical Novelty And Significance:** 3
**Empirical Novelty And Significance:** 3
**Recommendation:** 5
**Confidence:** 4

**Main Review:**

The authors have presented interesting results. The discrepancy between forward spike activation function and backward surrogate gradient function during backprop restricts training capability in SNNs. Modern deep learning relied on tricks like ReLU, initialization to balance forward/backward variance of activations (Glorot/He), skip connections etc. to overcome the vanishing gradient problem. The weight initialization approach in this work aims to do similar things.

While the authors have compared with recent works, I would like to bring the attention of authors to recent works from a group that authors have cited in their work, that use batch norm and threshold initialization as a way to mitigate the gradient training issue. I feel these two works are closely related to the authors work and both yield better performance results with very interesting SNN dynamics advantages like low latency etc. Since the authors have missed out comparing their work to these to works which are more in line with what authors are trying to do, I am a little concerned about the sanity and the novelty of this work.

[R1] Kim, Y., & Panda, P. (2020). Revisiting batch normalization for training low-latency deep spiking neural networks from scratch. arXiv preprint arXiv:2010.01729.
[R2] Kim, Y., & Panda, P. (2021). Optimizing deeper Spiking Neural Networks for Dynamic Vision Sensing. Neural Networks.

In [R1], Kim et al. present a temporal batchnorm technique to improve the gradient vanishing/explosion. I feel this work is very related to [R1]. So, it will be prudent for the authors to comment and show a comparsion. Further, in [R1] , the authors are able to train large-scale datasets from scratch like CIFAR100, TinyImagenet with a very low latency on rate-coded inputs. I am wondering if the author's method is scalable to non-trivial datasets beyond CIFAR10, MNIST (which are easy to get good accuracy with).

Based on this, my next question is, can authors comment how weight initialization impacts the overall spike activity, latency of processing of the network.

In [R2], Kim et al. propose a threshold initialization technique to improve the trainign of SNNs for DVS datasets and they looked at interesting datasets like N-Caltech (more complex than N-MNIST) as well as DVS-CIFAR10, DHP etc. Again, since the authors have looked into small scale datasets, it makes me question the scalability of their work. I think the authors should comment on this and also highlight how their work is better or orthogonal or complementary to these two works.




**Summary Of The Paper:**

This paper enhances the efficient BPTT training of SNNs by proposing an initialization method to match the response of spiking neurons in initial training. Their method bridges the spiking neuron response to the wisdom of traditional deep learning training, which may have an influence on future research like ANN-to-SNN conversions with LIF neurons or other SNN training methods. The authors conduct experiments on CIFAR10, MNIST, and neuromorphic datasets to show the efficacy of their technqiue. One of the main contribution is their theorteical analysis of iterative systems to model first-order integrate-and-fire neurons and investigate the response curve.


**Summary Of The Review:**

The results are interesting. I feel the authors should make a more thorough investigation of recent work that highlight similar ideas and yield better results on more complex and large-scale datasets. I am giving a rating of 5, but I am willing to chnage my score if the authors can convince me of the scalability and the novelty of their approach.

---

> ### Author Response · Authors · 2021-11-18
> **Response to Reviewer xJ5p (Part 1/2)**
>
> Thanks for your very detailed comments and suggestions for improvement. We would like to address your concerns below.
>
> > Since the authors have missed out comparing their work to these to works which are more in line with what authors are trying to do, I am a little concerned about the sanity and the novelty of this work.
>
> Thanks for pointing out these two works on SNN learning: `[Kim & Panda, 2021]` and `[Kim & Panda, 2020]`. They have brought very good supplements to our related work, which we have included in our manuscript.
> From the perspective of motivation, we agree that these two works are similar to ours in some degrees. But the approaches are quite different. The two methods are discussed on the overall network and take effect during the complete training process: `[Kim & Panda, 2020]` improved Batch Normalization to the time dimension and evaluated its impact on the entire network; `[Kim & Panda, 2020]` proposed an unsupervised pre-training technique to enable the rates to meet the constraints of the entire network response. However, the considerations in `[Kim & Panda, 2021]` and `[Kim & Panda, 2020]` lack direct mathematical modeling and discussion of neuron response.
> In contrast, our work focuses more on analyzing the reasons and effects of early spike generation from the perspective of spiking neurons and gets a theoretical initialization method based on this. In fact, we believe that these work with similar motivations should be complementary because our focus is not exactly the same. Therefore our method does not seek to compare the performance with these two algorithms alone.
> The field of deep learning is very particular about model performance. The excessive attention to network performance will make us ignore some important but subtle issues: how can we enable more neuron parameters to be effectively trained, which is also our motivation. It is especially important to the field of SNN and neuroscience. We believe that our work on the theoretical response of first-order linear integrate-and-fire neurons will have a positive influence on this problem.
>
>
> > Based on this, my next question is, can authors comment how weight initialization impacts the overall spike activity, latency of processing of the network.
>
> We claim that weight initialization affects the overall spike activity in a very limited way, as this effect has a short effective time and can be affected by the learning algorithm. The impact of our method on the overall spike activity is stated in Sec. 3.
> After obtaining the theoretical asymptotes, the analysis of the spiking model is simplified to a discussion in the linear space. Our parameter initialization method has two folds, each of which affects the spike activity:
> - First, we set the initial biases to $\frac{\theta(1-k)}{2\lambda}$ to maintain the symmetric distribution of spikes. They turn the weighted output of the original non-zero-mean distribution into a symmetric zero-mean current output (see Eq. 9 in the manuscript).
> - Second, we set the initial weights by a Gaussian distribution $\mathcal{N} ( 0, \frac{2\theta^2}{\lambda^2 n_{l-1}})$ because this setting can make the ratio of the variances of the currents between the layers stable (see Eq. 10 in the manuscript).
>
> Under such initialization, each layer will have a theoretical zero mean current output. In addition, the current distribution on the positive half axis will bring about a non-zero spike response, which helps SNNs generate gradients.
> To address your concern, we have modified the content in Sec. 3 to clarify the impact of initialization on spike activation. At the same time, we add a figure to further illustrate the gap between our initialization method and other initialization methods in the overall spike activity. Our initialization method can trigger spikes more effectively throughout the network compared to the other methods.

---

> ### Author Response · Authors · 2021-11-18
> **Response to Reviewer xJ5p (Part 2/2)**
>
> > Again, since the authors have looked into small scale datasets, it makes me question the scalability of their work. I think the authors should comment on this and also highlight how their work is better or orthogonal or complementary to these two works.
>
> Thank you for your insightful feedback about scalability. We added the experiments on CIFAR100 and TinyImageNet. We used the VGG11 network from `[Kim & Panda, 2020]` and used constant coding to train these two datasets. See Appendix for details. The experimental results show that our initialization can successfully train the network in more neuron parameter settings. However, the accuracy of such training is still far from the best performance of these two datasets.
> At the same time, we have also added various initialization methods to the network using BN and BNTT and trained them to test performance. The experimental results show that the use of BNTT/BN makes the effect of initialization insignificant. We think it is related to the fact that BN/BNTT affects the gradient update in the entire training procedure.
> These facts make us aware of two problems with our method:
>
> - Our method does not guarantee a better generalization performance on complex datasets.
> - The performance benefits of the initialization methods for complex datasets are not high with BN.
>
> Therefore, we are not stating that our initialization method can outperform some well-designed learning algorithms. Instead, we believe our method is a useful supplement to make SNNs trainable on larger datasets.
> We find out that our method has limitations on complex datasets. The dimensionality of the input data of the complex dataset increases, resulting in an increase in current noise due to spike quantization. Therefore, we added the discussion of the limitations of this work in Sec. 7. The analysis in Sec. 2.2 may need to quantify the realistic spike noise if our method needs to adapt to complex datasets. We think there is still a lot of work remaining. Further extensions to complex datasets will be our future work.
>
> *[Kim & Panda, 2021] Kim, Y., & Panda, P. (2021). Optimizing Deeper Spiking Neural Networks for Dynamic Vision Sensing. Neural Networks, 144, 686-698.*
>
> *[Kim & Panda, 2020] Kim, Y., & Panda, P. (2020). Revisiting batch normalization for training low-latency deep spiking neural networks from scratch. arXiv preprint arXiv:2010.01729.*

---

### Official Review · Reviewer_gWVF · 2021-11-04

**Correctness:** 3
**Technical Novelty And Significance:** 3
**Empirical Novelty And Significance:** 3
**Recommendation:** 6
**Confidence:** 4

**Main Review:**

The paper addresses an important problem of inefficiencies in trainign SNN models and presents a compelling argument for the inefficient SNN initialization and demonstrate that improving it so that neurons are responding from training onset can increase both convergence and test accuracy.

While the paper is technically sound and offers a substantial result, the presentation is quite  convoluted and could be substantially streamlined and would benefit dramatically from a read by someone fluent in English. Just to mention a few issues, the introduction section is overly verbose and does not seem to be always to the point. Many statements are non-informative, like "neurons respond in a suitable region..." or "producing proper amount of spike" where "suitable region" and "proper amount" is never defined. Incorrect use of terms abound (non-differential => non-differentiable etc) that can seriously obscure the authors' intentions.

Minor issues:
Figure 2: Dashed lines referenced but not displayed; what is the difference; why the actual neuron responses (circles) are not shown for the full range of inputs?




**Summary Of The Paper:**

The authors tackle the problem of improving spiking neural net training with better initialization. They observe that error backpropagation efficiency in SNNs depends on availability of responding neurons and show that traditional surrogate backprop methods are inefficient as they take many iterations to move weights to the regime supporting neuronal responses.
The authors approximate the SNN neuron model (LIF, IF etc) I/O function using a piecewise-linear  iterative expression and show that clipping the output to (0,1) and normalizing the variance of the random weight initialization to insure that neurons produce substantial responses even with initial random weights results improves SNN training substantially. The improvement is to a large extent due to early onset of convergence. The authors demonstrate the superiority of their approach on MNIST, N-MNIST, DVS-MNIST and CIFAT datasets and present extensive experiments using different optimization algorithms and response functions.


**Summary Of The Review:**

While the paper presents an important and technically sound contribution, the quality of presentation dramatically reduces the value of the manuscript.

---

> ### Author Response · Authors · 2021-11-18
> **Response to Reviewer gWVF**
>
> Thank you for your thoughtful comments and suggestions for improvement. We are glad you felt that our work is important and technically sound. We would like to address your concerns below.
>
> > While the paper is technically sound and offers a substantial result, the presentation is quite convoluted and could be substantially streamlined and would benefit dramatically from a read by someone fluent in English.
>
> Thank you for your suggestions. We have proofread our manuscript. Please refer to general response 1 for more detail.
>
> > Minor issues: Figure 2: Dashed lines referenced but not displayed; what is the difference; why the actual neuron responses (circles) are not shown for the full range of inputs?
>
> Here, we give a detailed explanation of Figure 2. Figure 2 was intended to illustrate the consistency of the theoretical response curve and the actual response observation. The slant asymptote lines of the theoretical response curve are displayed in Figure 2 as green dashed lines. Actual neuron responses plotted in the figure were recorded from a layerwise information transmission simulation: The input was randomly sampled and encoded as Poisson spike trains, which were then fed into a linear layer to produce current input. Then the current input was fed into the neuron model and triggered spikes. Finally, we averaged the current input and corresponding spike output, and plotted the pairs on the figure. As the simulated actual neuron responses can only reach a certain range, the actual neuron responses are not shown for the full range. To address your concern, we have changed the simulated current input to the perturbed input of zero-mean noise and replotted Figure 2. Note that the shape of the actual responses in the subplot (a) in the revised figure is like stairs. A similar stair-like figure can also be found in `[Han et al., 2020]`, which described the same asymptote in the context of ANN-SNN conversion. Accordingly, we alter the statement of "It can be seen that the actual neuron responses are distributed almost along a continuous line" to a more strict one "The actual neuron responses match the theoretical $f-i$ curve for most inputs".
>
> *[Han et al., 2020] Han, B., Srinivasan, G., & Roy, K. (2020). RMP-SNN: Residual membrane potential neuron for enabling deeper high-accuracy and low-latency spiking neural network. In Proceedings of the IEEE/CVF Conference on Computer Vision and Pattern Recognition (pp. 13558-13567).*

---

### Author Response · Authors · 2021-11-18
**To All Reviewers**

We thank all reviewers for their time and insightful feedback. Here we enumerate the common concerns raised by the reviewers and give general responses:

## General Response 1: Improvements on language, structure, and terms

We have made the following adjustments in this regard.
- In terms of English expression, we recognized our limitations in the correct use of English, so we thoroughly proofread and edited our manuscripts with the help of an English proofreading agency.
- In the article structure, we condensed the content to make the article more to the point.
- In terms of rigorousness, we revised those non-informative expressions and corrected the use of terms.



## General Response 2: Experiments on complex datasets and large models

We have conducted experiments on CIFAR100 and TinyImageNet. We used the VGG11 network from `[Kim & Panda, 2020]` and used constant coding to train these two datasets. Please see Appendix A.3 for the details. While the experimental results show that using our initialization can successfully train the network in more neuron parameter settings (Table A8), the accuracy of such training on larger datasets still needs to be improved with further exploration.
We have added the discussion on limitations in Sec. 7. For complex datasets, there is an increase in the current noise due to spike quantization. And this will result in a shift in the theoretical $f-i$ curve `[Gerstner et al., 2014]`. The analysis in Sec. 2.2 may need to quantify the realistic spike noise if our method needs to adapt to complex datasets. We think there is still a lot of work remaining, which is difficult to be fully explained in an article of 9 pages. We are glad to investigate it further and extend it to complex datasets in future work.

*[Gerstner et al., 2014] Gerstner, W., Kistler, W. M., Naud, R., & Paninski, L. (2014). Neuronal dynamics: From single neurons to networks and models of cognition. Cambridge University Press.*

*[Kim & Panda, 2020] Kim, Y., & Panda, P. (2020). Revisiting batch normalization for training low-latency deep spiking neural networks from scratch. arXiv preprint arXiv:2010.01729.*

Apart from that, we would like to point out other major improvements we have made in our paper:
- We added a figure in Sec. 3 to illustrate the impact of initialization methods on the overall spike activity.
- We added a new section named "Discussions and Limitations" before "Conclusion" and included the discussion of limitations in General Response 2.
- We added work with similar motivations to our work in related work and condensed the original content there.

---

### Decision · Program_Chairs · 2022-01-20

**Decision:**

Reject

**Comment:**

The paper derives a new parameter initialization for deep spiking neural networks to overcome the vanishing gradient problem.

During the review, concerns were expressed about how well the method would scale to larger neural networks. It was also questioned how this parameter initialization technique compares with a recently proposed batch normalization technique, especially when training larger neural network on more challenging datasets. There were also concerns raised about the readability of the paper.

I commend the authors for improving the readability of their paper in their revision. I also commend them for taking the time to implement the comparisons requested by the reviewers. These new comparisons revealed that batch normalization and its recently proposed variant were superior to the initialization method on its own, and that the initialization proposed in the paper did not significantly improve performance when paired with batch norm [[1](https://openreview.net/forum?id=T8BnDXDTcFZ&noteId=yIAPcSbUAQ0)]. The authors also acknowledged based on the new results, that their proposed parameter initialization scheme appears to fail to scale to more complex datasets and networks, especially relative to competing methods, which invalidates a key claim that their approach can "accelerate training and get better accuracy compared with existing methods" [[2](https://openreview.net/forum?id=T8BnDXDTcFZ&noteId=j12fwayWEb)].

The recommendation is to reject the paper in its current form.